# Ancient origins of arthropod moulting pathway components

**André Luiz de Oliveira, Andrew Calcino, Andreas Wanninger***

Department of Integrative Zoology, Faculty of Life Sciences, University of Vienna, Vienna, Austria

**Abstract** Ecdysis (moulting) is the defining character of Ecdysoza (arthropods, nematodes and related phyla). Despite superficial similarities, the signalling cascade underlying moulting differs between Panarthropoda and the remaining ecdysozoans. Here, we reconstruct the evolution of major components of the ecdysis pathway. Its key elements evolved much earlier than previously thought and are present in non-moulting lophotrochozoans and deuterostomes. Eclosion hormone (EH) and bursicon originated prior to the cnidarian-bilaterian split, whereas ecdysis-triggering hormone (ETH) and crustacean cardioactive peptide (CCAP) evolved in the bilaterian last common ancestor (LCA). Identification of EH, CCAP and bursicon in Onychophora and EH, ETH and CCAP in Tardigrada suggests that the pathway was present in the panarthropod LCA. Trunk, an ancient extracellular signalling molecule and a well-established paralog of the insect peptide prothoracicotropic hormone (PTTH), is present in the non-bilaterian ctenophore *Mnemiopsis leidyi*. This constitutes the first case of a ctenophore signalling peptide with homology to a neuropeptide.
DOI: https://doi.org/10.7554/eLife.46113.001

## Introduction

Ecdysis or moulting, which describes the process of shedding the outer integument, the cuticle, is a defining feature of Ecdysozoa (arthropods, tardigrades, onychophorans, nematodes and related phyla) (*Aguinaldo et al., 1997*; *Schmidt-Rhaesa et al., 1998*; *de Rosa et al., 1999*; *Dunn et al., 2008*; *Telford et al., 2008*). Despite superficial similarities of the 'moulting behaviour' within Ecdysozoa, the neuroendocrine components underlying this process remain elusive for the majority of the ecdysozoans outside of Arthropoda. This includes well-established model organisms such as the nematode *Caenorhabditis elegans*, for which the gene regulatory network responsible for ecdysis remains to be fully resolved (*Frand et al., 2005*; reviewed by *Page et al., 2014* and *Lažetić and Fay, 2017*).

In arthropods, ecdysis can be divided into three distinct stages, pre-ecdysis, ecdysis and post-ecdysis. Each of these stages correlates with major behavioural, molecular and cellular changes and encompasses a series of specific muscular contractions controlled by a cascade of hormones and neuropeptides (*Truman, 2005*). Studies in insects have revealed that the major components of this peptidergic signalling pathway are ecdysis-triggering hormone (ETH), eclosion hormone (EH), crustacean cardioactive peptide (CCAP) and bursicon (*Gammie and Truman, 1997a*; *Gammie and Truman, 1997b*; *Zitnan et al., 1999*; *Clark et al., 2004*; *Kim et al., 2006a*; *Kim et al., 2006b*; *Arakane et al., 2008*; *Lee et al., 2013*). The process begins with the release of prothoracicotropic hormone (PTTH) from neurohemal organs. PTTH initiates a signalling cascade that results in the biosynthesis of ecdysteroids (i.e., steroid hormones synthesised from ingested cholesterol), including ecdysone (E) and 20-hydroxyecdysone (20E) (*Figure 1*). The decline of the ecdysone titre due to the ecdysone-inactivating enzyme cytochrome P450 protein Cyp18a1 (*Guittard et al., 2011*; reviewed by *Rewitz et al., 2013*) triggers the release of ETH that, in turn, causes the release of EH. These two hormones mutually enhance one another in a positive feedback loop to control and regulate pre-

*For correspondence:
andreas.wanninger@univie.ac.at

**Competing interests:** The authors declare that no competing interests exist.

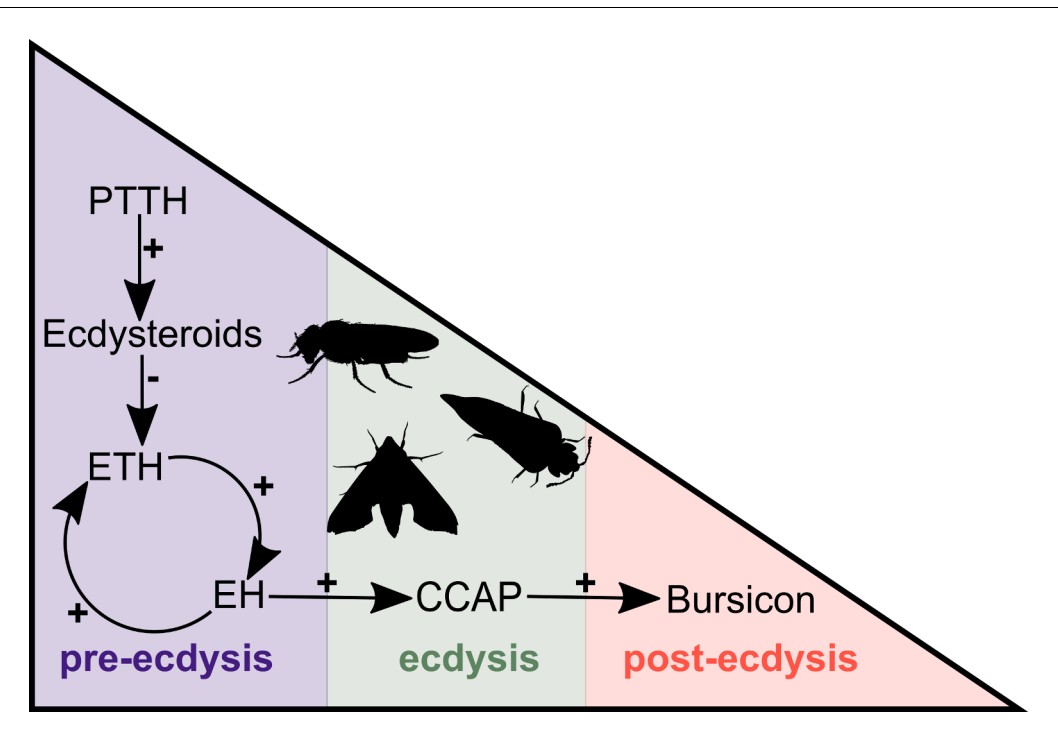

**Figure 1.** Simplified overview of the neuropeptide/hormone signalling pathway at moulting. PTTH initiates a signalling cascade that results in the biosynthesis of ecdysone. The decline of the ecdysone titre triggers the release of ETH that, in turn, causes the release of EH. These two hormones mutually enhance one another in a positive feedback loop to control and regulate pre-ecdysis behaviour. With the ensuing release of CCAP, caused by EH, pre-ecdysis ceases and the ecdysis motor program is started. Finally, bursicon responds to the increasing levels of CCAP and initiates post-ecdysis behaviour and cuticle tanning. This figure is based on the studies of *McNabb et al. (1997)* and *Clark et al. (2004)*. Animal silhouettes were obtained under Public Domain licence at phylopic (http://phylopic.org/), unless otherwise indicated. Beetle: T. Michael Keesey after Ponomarenko (available for reuse under https://creativecommons.org/publicdomain/zero/1.0/); moth: by Gareth Monger (available for reuse under https://creativecommons.org/licenses/by/3.0/); *Drosophila*: Thomas Hegna (available for reuse under https://creativecommons.org/publicdomain/zero/1.0/).
DOI: https://doi.org/10.7554/eLife.46113.003

ecdysis behaviour (*Figure 1*). With the ensuing release of CCAP, caused by EH, pre-ecdysis ceases and the ecdysis motor program is initiated. Finally, bursicon responds to the increasing levels of CCAP and initiates post-ecdysis behaviour and cuticle tanning (*Figure 1*).

Comparative biochemical, genomic and transcriptomic analyses revealed that ecdysteroids and the required genes responsible for their biosynthesis are present outside of Ecdysozoa, showing that some key molecular players of moulting predate the origin of Ecdysozoa (*Mendis et al., 1984*; *Nolte et al., 1986*; *Garcia et al., 1989*; *Barker et al., 1990*; *Schumann et al., 2018*). Such integrative and comparative analyses have so far not been conducted on the major components of the peptidergic signalling system underlying moulting. To fill this gap in knowledge, we explored the distribution of PTTH, ETH, EH, CCAP and bursicon ligand-receptor pairs across Metazoa.

## Results and discussion

PTTH is a neurohormone with a proposed origin at the base of Arthropoda that is believed to have evolved from the duplication of the ancient and widely distributed bilaterian signalling molecule-encoding gene *trunk* (*Rewitz et al., 2009*; *Jékely, 2013*). By screening 39 metazoan genomes and 57 transcriptomes (*Supplementary file 1*), we found that the PTTH peptide is present in *Drosophila* and *Tribolium* but absent in the house spider *Parasteatoda tepidariorum* and the crustacean

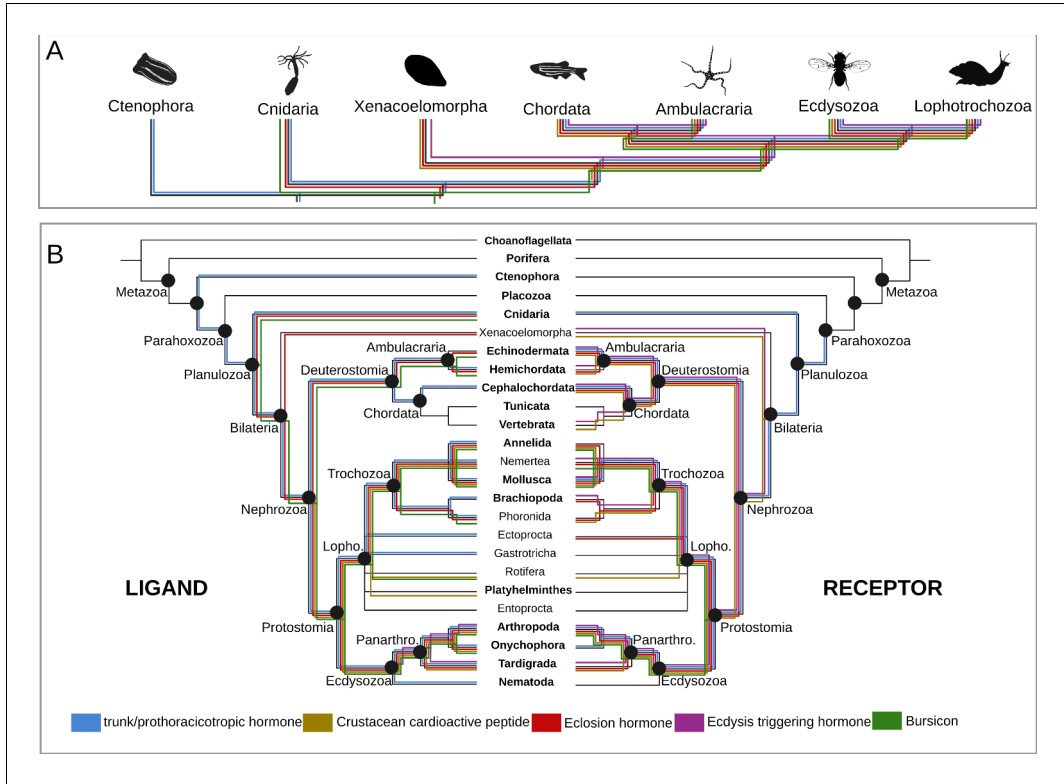

**Figure 2.** Origin and distribution of the key ligand-receptor components of the arthropod moulting signalling pathway across Metazoa. (**A**) Simplified phylogeny (based on *Dunn et al., 2014*) of Metazoa showing the lineages in which the key components of the arthropod moulting signalling pathway are present. Note that Porifera and Placozoa, that lack the moulting pathway components investigated here, are omitted for clarity. Coloured lines indicate the presence of a given ligand and/or receptor in a given lineage. Eclosion hormone and bursicon peptidergic systems originated prior to the cnidarian-bilaterian split, whereas the ecdysis-triggering hormone and crustacean cardioactive peptide trace back to the last common ancestor of Bilateria. PTTH is an insect-specific neuropeptide. (**B**) Expanded phylogeny of Metazoa with Porifera as the earliest branching clade (adapted from *Dunn et al., 2014*). Coloured lines indicate the presence of a given ligand (right side) and receptor (left side) in a given lineage. Phylum name in bold indicates the availability of genomic data. Note that although the *trunk* ortholog was not retrieved from the genomes of *Nematostella vectensis* and *Caenorhabditis elegans*, similarity searches against publicly available protein databases identified this gene in other cnidarian and nematode species. Animal silhouettes were obtained under Public Domain licence at phylopic (http://phylopic.org/), unless otherwise indicated. Credited images: Ctenophora: Martini (available for reuse under https://creativecommons.org/publicdomain/zero/1.0/); Cnidaria: Jack Warner (available for reuse under https://creativecommons.org/publicdomain/zero/1.0/); Xenacoelomorpha: Andreas Hejnol (available for reuse under https://creativecommons.org/licenses/by-nc/3.0/); Chordata: Jake Warner (available for reuse under https://creativecommons.org/publicdomain/zero/1.0/); Ambulacraria: Noah Schlottman (photograph from Casey Dunn available for reuse under https://creativecommons.org/licenses/by-sa/3.0/); Ecdysozoa: Thomas Hegna based on picture by Nicolas Gompel (available for reuse under https://creativecommons.org/publicdomain/mark/1.0/); Lophotrochozoa: Fernando Carezzano (available for reuse under https://creativecommons.org/publicdomain/zero/1.0/).
DOI: https://doi.org/10.7554/eLife.46113.004

The following source data and figure supplements are available for figure 2:

**Source data 1.** PTTH/trunk/torso proteins and tree associated files.
DOI: https://doi.org/10.7554/eLife.46113.011
**Source data 2.** ETH/ETH-receptor proteins and tree associated files.
DOI: https://doi.org/10.7554/eLife.46113.012
**Source data 3.** EH/EH-receptor proteins and tree associated files.
DOI: https://doi.org/10.7554/eLife.46113.013
**2 - Source data 4** CCAP/CCAP-receptor proteins and associated tree files.
DOI: https://doi.org/10.7554/eLife.46113.014

*Figure 2 continued on next page*

*Figure 2 continued*

**Source data 5.** Bursicon/rickets protein and tree associated files.
DOI: https://doi.org/10.7554/eLife.46113.015

**Figure supplement 1.** 2D cluster maps of trunk/PTTH, EH, CCAP and bursicon ligands reflecting the evolutionary relatedness of the key arthropod moulting components among metazoans.
DOI: https://doi.org/10.7554/eLife.46113.005
**Figure supplement 2.** Phylogenetic analysis of the PTTH/trunk receptor tyrosine kinase torso showing the presence of torso receptor in cnidarians, lophotrochozoans, ecdysozoans and deuterostomes.
DOI: https://doi.org/10.7554/eLife.46113.006
**Figure supplement 3.** Phylogenetic analysis of the ecdysis-triggering hormone receptor showing the presence of ETH-receptor in bilaterians.
DOI: https://doi.org/10.7554/eLife.46113.007
**Figure supplement 4.** Phylogenetic analysis of the guanylyl cyclase eclosion hormone receptor showing the presence of EH-receptor in ecdysozoans, lophotrochozoans, ambulacrarians and cephalochordates.
DOI: https://doi.org/10.7554/eLife.46113.008
**Figure supplement 5.** Phylogenetic analysis of the G protein-coupled CCAP receptor showing the presence of CCAP-receptor in ecdyzosoans, lophotrochozoans, deuterostomes (including vertebrates) and acoels.
DOI: https://doi.org/10.7554/eLife.46113.009
**Figure supplement 6.** Phylogenetic analysis of the bursicon G protein-coupled receptor rickets showing the presence of rickets receptor in arthropods and lophotrochozoans.
DOI: https://doi.org/10.7554/eLife.46113.010

*Parhyale hawaiensis*, suggesting that PTTH is an insect innovation (*Figure 2A,B*, *Figure 2—figure supplement 1A*, *Figure 2—source data 1*). The ablation of PTTH-producing neurons in *Drosophila* generates an imbalance in ecdysone biosynthesis, causing developmental delay, prolonged duration of feeding and larger individuals with reduced fecundity (*McBrayer et al., 2007*). These findings indicate that PTTH, at least in *Drosophila*, regulates developmental timing and body size, but is not essential for moulting. *Trunk*, the paralog of *ptth*, has previously been identified in arthropods, annelids, mollusks and the cephalochordate *Branchiostoma floridae* (*Jékely, 2013*). Our study expands the phyletic distribution of *trunk* to onychophorans, tardigrades, gastrotrichs, brachiopods, nemerteans, ectoprocts, phoronids and hemichordates (*Figure 2A,B*, *Figure 2—figure supplement 1A*, *Figure 2—source data 1*). Although we did not find a *trunk* ortholog in the genome of the sea anemone *Nematostella vectensis*, our similarity searches against the NCBI protein database led to the identification of this protein in other anthozoans, namely *Stylophora pistillata* (PFX31008.1) and *Orbicella faveolata* (XP_020630744.1 and XP_020630745.1). More importantly, our multi-species screen also recovered a trunk-like peptide in the ctenophore *Mnemiopsis leidyi* with high similarity (p-value < 1e-05) to lophotrochozoan, deuterostome and ecdysozoan trunk sequences (*Figure 2—figure supplement 1A*; *Figure 3A,B*). By similarity-based clustering we were able to demonstrate homology of the ctenophore trunk-like peptide with the insect *trunk* paralog, *ptth* (*Figure 3A*, *Figure 3—source data 1*; see also *Rewitz et al., 2009*; *Jékely, 2013*). This extends the phyletic distribution of trunk to the ctenophores (see, e.g., *Halanych, 2004*; *Dunn et al., 2008*; *Moroz et al., 2014*; *Jékely et al., 2015*; *Pisani et al., 2015* for discussion).

PTTH and trunk share a common receptor, the tyrosine kinase torso (*Rewitz et al., 2009*). Similar to its ligands, torso (*Rewitz et al., 2009*) proved also to be much more ancient than commonly assumed. We identified torso sequences in deuterostomes, lophotrochozoans, cnidarians and ecdysozoans (*Figure 2—figure supplement 2*), indicating that the trunk-torso neuropeptide signalling pathway dates back at least as far as the last common ancestor of Cnidaria, Ctenophora and Bilateria and is thus not restricted to Bilateria as suggested previously (e.g., *Jékely, 2013*) (*Figures 2A,B* and *4A*).

In insects, the first hormone released in response to decreasing ecdysone levels is usually ETH (*Zitnan et al., 1996*; *Zitnan et al., 1999*) although in the lepidopteran *Manduca sexta* the neuropeptide corozanin acts as the trigger for the release of ETH from the epitracheal glands (*Kim et al., 2004*). Knockdown of the *eth* gene in *Drosophila* (*Park et al., 2002*) and of *eth* and its receptors in *Tribolium* and *Schistocerca* (*Arakane et al., 2008*; *Lenaerts et al., 2017*) lead to lethality at the expected onset of ecdysis, demonstrating the essential role of the ETH peptidergic signalling system in moulting (*Park et al., 2002*; *Arakane et al., 2008*; *Lenaerts et al., 2017*; *Shi et al., 2017*). Our

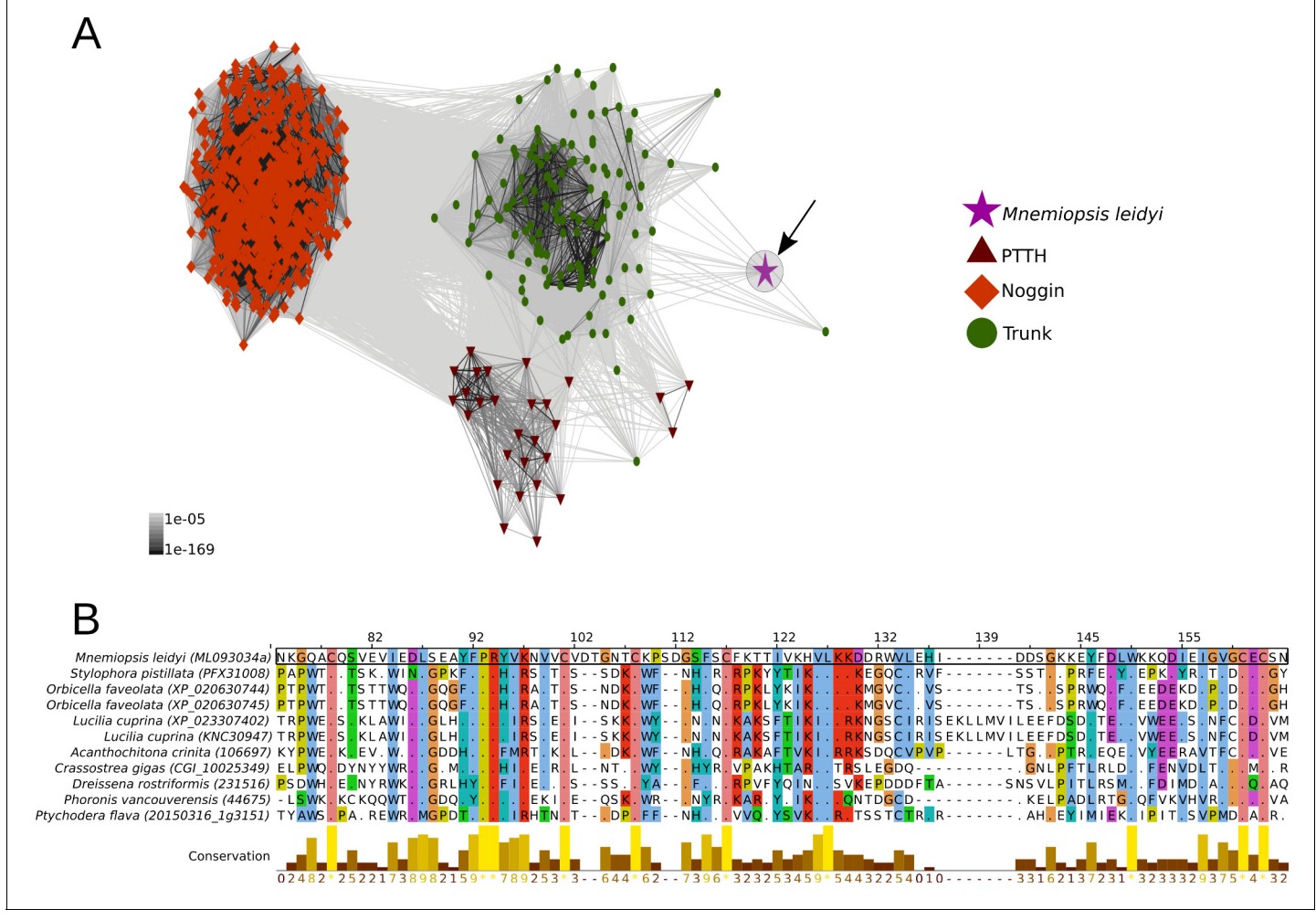

**Figure 3.** Cluster analysis of *prothoracicotropic hormone (ptth)*, *trunk*, *noggin* orthologs and multiple sequence alignment of the ctenophore trunk-like peptide and the metazoan ortholog sequences. (A) 2D cluster map of *ptth*, *trunk* and *noggin* genes. Red triangles correspond to *ptth* homologs, green parallelograms correspond to *noggin* homologs and red circles correspond to *trunk* homologs. The ctenophore *trunk* gene sequence is represented by the pink star. Edges represent BLAST connections of P value > 1e-05. Note that the ctenophore trunk peptide is indirectly connected to insect PTTH sequences via transitive BLAST connections. (B) Multiple sequence alignment representation of ctenophore trunk sequence and its metazoan orthologs produced by Jalview 2 (*Waterhouse et al., 2009*). Only the sequences directly connected to the ctenophore sequence in the 2D cluster map are included in the multiple sequence alignment. The conservation histogram corresponds to the number of conserved amino acid physico-chemical properties for each column of the alignment.

DOI: https://doi.org/10.7554/eLife.46113.016

The following source data is available for figure 3:

**Source data 1.** Ctenophore trunk cluster peptide map.
DOI: https://doi.org/10.7554/eLife.46113.017

screening and phylogenetic analyses confirmed the presence of ETH and its receptor in tardigrades and arthropods, thus corroborating previous studies (*Figure 2B*, *Figure 2—figure supplement 3*, *Figure 2—source data 2*) (*Zitnan et al., 1996*; *Zitnan et al., 1999*; *Park et al., 2002*; *Arakane et al., 2008*; *Veenstra et al., 2012*; *Lenaerts et al., 2017*; *Koziol, 2018*; *Zhu et al., 2019*). For Arthropoda, the ETH ligand was only found in insects (*Drosophila* and *Tribolium*), but was lacking in the crustacean *Parhyale hawaiensis* and the arachnid *Parasteatoda tepidariorum*. However, studies on the two mites Panonychus citri and Tetranychus urticae as well as several decapods have shown the presence of the ETH ligand in chelicerates and crustaceans (in which the homology was reconfirmed by our clustering analysis) (*Veenstra et al., 2012*; *Veenstra, 2016*; *Zhu et al., 2019*). Surprisingly, we did not find the entire ETH signalling pathway in the two onychophoran genomes

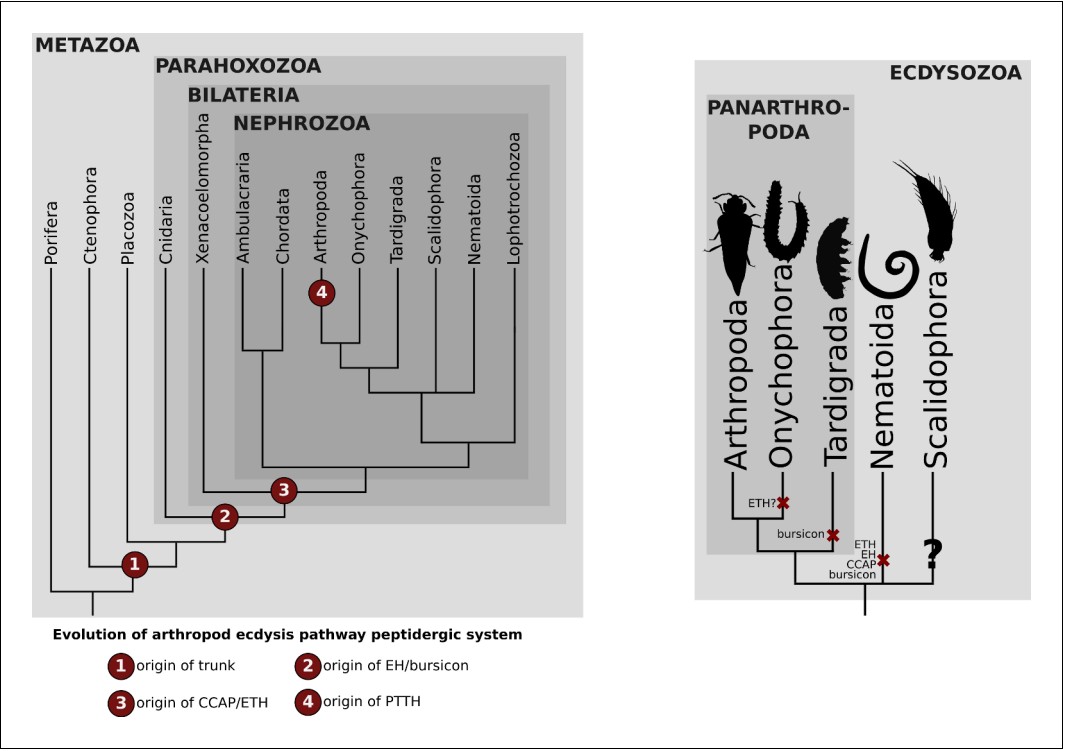

**Figure 4.** Distribution of the arthropod peptidergic system components throughout Metazoa. (**A**) Simplified phylogeny of Metazoa with Porifera as the most basally branching clade (adapted from *Dunn et al., 2014*) showing the origin of the trunk/PTTH, eclosion-hormone (EH), bursicon, crustacean cardioactive peptide (CCAP) and ecdysis-triggering hormone (ETH) peptigergic systems. (**B**) Distribution of the arthropod peptigerdic system components within Panarthropoda. Secondary losses are depicted by the red crosses followed by the name of the peptide system absent in the lineage. Note that ETH and bursicon, two vital components underlying moulting in insects, were possibly secondarily lost in the Onychophora and Tardigrada (indicated by the red cross), respectively. Genomic and transcriptomic homology searches within the Kinorhyncha, Priapulida and Loricifera (condensed into the clade Scalidophora in *Figure 1B*) were not performed in this study (indicated by the question mark). Animal silhouettes were obtained under Public Domain licence at phylopic (http://phylopic.org/), unless otherwise indicated. Arthropoda: T. Michael Keesey after Ponomarenko (available for reuse under https://creativecommons.org/publicdomain/zero/1.0/); Onychophora: Noah Schlottman, photo by Adam G. Clause (available for reuse under https://creativecommons.org/licenses/by-sa/3.0/); Tardigrada: Fernando Carezzano (available for reuse under https://creativecommons.org/publicdomain/zero/1.0/); Nematoida: Mali'o Kodis, image from the Smithsonian Institution (available for reuse under https://creativecommons.org/licenses/by-nc-sa/3.0/); Scalidophora: Noah Schlottman, photo by Martin V. Sørensen (available for reuse under https://creativecommons.org/licenses/by-sa/3.0/).
DOI: https://doi.org/10.7554/eLife.46113.018

analysed herein (*Figures 2B* and *4B*, *Figure 2—figure supplement 3*, *Figure 2—source data 2*). Due to the fragmented nature of the onychophoran genomes a final statement whether this signalling system was indeed lost in this lineage cannot be made at present.

No *eth* ortholog was identified outside the panarthropods, including the nematode *C. elegans*, suggesting that this gene originated in the last common ancestor of Panarthropoda (*Figure 2B*). Our findings on the distribution of the ETH receptor are in agreement with the results of previous studies and demonstrate the presence of this receptor in arthropods (insects, crustaceans and arachnids), mollusks, nemerteans, brachiopods, echinoderms, cephalochordates (in which we found a substantial expansion of *eth-receptor* homologs in the genomes of *Branchiostoma floridae* and *B. belcheri*), vertebrates and acoels (*Park et al., 2003*; *Roller et al., 2010*; *Veenstra et al., 2012*; *Mirabeau and Joly, 2013*; *Lenaerts et al., 2017*; *Thiel et al., 2018*; *Zhu et al., 2019*) (*Figures 2B* and *4A*, *Figure 2—figure supplement 3*, *Figure 2—source data 2*). This provides evidence for the

presence of individual components of the ETH signalling system already at the base of Bilateria (*Figures 2A* and *4A*).

Eclosion hormone (EH) was first identified as a blood-borne factor (*Truman and Riddiford, 1970*) in three lepidopteran species, *Hyalophora cecropia*, *Antheraea polyphemus* and *Antheraea pernyi*. A positive feedback loop between EH and ETH was found in *Manduca* and suggested in *Tribolium* (*Ewer et al., 1997*; *Arakane et al., 2008*), whereas in *Drosophila*, EH has been described either acting downstream of ETH (*Kim et al., 2006a*; *Kim et al., 2006b*) or in a positive endocrine feedback loop with ETH (*Krüger et al., 2015*). Despite EH being a key regulator of ecdysis in insects, *eh* knockout in *Drosophila melanogaster* did not abolish ecdysis, but instead produced flies with discrete behavioural deficits such as slow and uncoordinated eclosion. This shows that EH is involved, but does not play an essential role, in moulting in these insects (*McNabb et al., 1997*). This result, however, has been contested in a more recent study using *eh* null mutants in *Drosophila* (*Krüger et al., 2015*), showing that the lack of *eh* function is lethal during larval fruit fly ecdysis.

Traditionally considered to be confined to arthropods, recent studies showed the presence of EH and its receptor, a guanylyl cyclase, in echinoderms and tardigrades (*Zandawala et al., 2017*; *Koziol, 2018*). Our study corroborates these findings but considerably expands the presence of the EH ligand to cnidarians, acoels, hemichordates, lophotrochozoans (mollusks, annelids, nemerteans and phoronids) and onychophorans (*Figure 2B*, *Figure 2—figure supplement 1B*, *Figure 2—source data 3*). All EH ligand orthologs harbour the six cysteine conserved residues (*Zitnan et al., 2007*) except for Cnidaria, in which only five are present. The identification of the EH receptor in ambulacrarians, mollusks, annelids, nemerteans and phoronids suggests co-evolution of this ligand-receptor pair throughout Metazoa (*Figure 2B*, *Figure 2—figure supplement 4*, *Figure 2—source data 3*). Although the *eh-receptor* gene was not found in Cnidaria, Xenacoelomorpha and Onychophora, its distribution includes the Brachiopoda, Ectoprocta and Cephalochordata lineages. Consequently, our findings shift the ancestry of this peptidergic pathway back to the cnidarian-bilaterian split (*Figures 2A* and *4A*).

First isolated from the shore crab *Carcinus maenas*, crustacean cardioactive peptide (CCAP) is a highly conserved amidated neuropeptide that increases heart rate in crustaceans and insects (*Stangier et al., 1987*; *Cheung et al., 1992*; *Lehman et al., 1993*; *Suggs et al., 2016*). CCAP has multiple functions in addition to its cardioacceleratory activity, such as accelerating the frequency and amplitude of oviduct contractions in the locust *Locusta migratoria* (*Donini et al., 2001*) and regulating the release of digestive enzymes in the cockroach *Periplaneta americana* (*Sakai et al., 2006*). CCAP is important for ecdysis in crustaceans and insects where it initiates the stereotyped sequence of behaviours that mark the end of the pre-ecdysis stage (*Gammie and Truman, 1997a*; *Gammie and Truman, 1997b*; *Phlippen et al., 2000*; *Arakane et al., 2008*; *Lee et al., 2013*). However, transgenic *Drosophila* larvae lacking CCAP neurons moult normally and only exhibit a prolonged pre-ecdysis behaviour (*Clark et al., 2004*).

Only three studies focusing on the CCAP signalling pathway components are available outside of Arthropoda. In the snail *Lymnaea stagnalis* (*Vehovszky et al., 2005*), immunostaining revealed a dense network of CCAP-positive fibres that likely function to regulate parts of the feeding behaviour. In the oyster *Saccostrea glomerata* (*In et al., 2016*) and in the cuttlefish *Sepia officinalis* (*Endress et al., 2018*), in vivo bioassays using synthesised neuropeptides and immunohistochemistry suggested that the CCAP signalling pathway is involved in reproduction (e.g., spawning, oocyte transport, egg-laying). Additionally, *Sepia* CCAP has been shown to increase the tonus of the vena cava, demonstrating its role in the regulation of hemolymph circulation (*Endress et al., 2018*). These results indicate that in both, mollusks and arthropods, CCAP functions in feeding, reproduction and regulation of hemolymph circulation, suggesting that these may have been its ancestral roles. In arthropods, co-option of CCAP into the ecdysis pathway expanded this set of functions to include moulting.

The CCAP receptor is a G protein-coupled receptor (GPCR) that was first described from the *Drosophila* genome and subsequently identified in many other insects (*Cazzamali et al., 2003*; *Arakane et al., 2008*; *Vogel et al., 2013*). We confirm here the presence of a CCAP ligand in mollusks, annelids, arthropods and tardigrades, as stated earlier (*Veenstra, 2010*; *Jékely, 2013*; *Mirabeau and Joly, 2013*; *Conzelmann et al., 2013*; *Stewart et al., 2014*; *Ahn et al., 2017*; *Zhang et al., 2018*; *Koziol, 2018*), and extend the distribution of the CCAP ligand to three

additional lophotrochozoan phyla (Nemertea, Platyhelminthes and Rotifera) as well as to the remaining panarthropod phylum Onychophora (*Figures 2B* and *4B*, *Figure 2—figure supplement 1C*, *Figure 2—source data 4*). Interestingly, the CCAP ligand is absent from all investigated deuterostome genomes analysed (*Figure 2—figure supplement 1C*, *Figure 2B*, *Figure 2—source data 4*). The *ccap-receptor* ortholog was found in acoels, lophotrochozoans and panarthropods except Onychophora (*Figure 2—figure supplement 5*). Surprisingly, we found the receptor also in all deuterostome phyla (Echinodermata, Hemichordata, Vertebrata, Cephalochordata) except Tunicata (*Figure 2—figure supplement 5*, *Figure 2—source data 4*). These results reinforce the suggested origin of the ligand-receptor pair at the base of Bilateria and points towards a possible loss of the CCAP ligand in the Deuterostomia lineage (*Figures 2B* and *4A*, *Figure 2—figure supplement 5*, *Figure 2—source data 4*).

Bursicon was identified as a neurohormone responsible for cuticle sclerotization and melanisation (tanning) during post-ecdysis (*Cottrell, 1962a*; *Cottrell, 1962b*; *Fraenkel and Hsiao, 1965*). Recent studies have shown that bursicon also has a mild effect on the regulation of pre-ecdysis and is important for the proper execution of post-ecdysis in *Manduca*, *Drosophila* and *Tribolium* as well as for the development of wings and other integumentary structures (*Baker and Truman, 2002*; *Dewey et al., 2004*; *Arakane et al., 2008*; *Bai and Palli, 2010*). Together with the ecdydis-triggering hormone signalling system, bursicon is an indispensable component of the moulting behaviour in insects (*Arakane et al., 2008*). Previous studies show that bursicon is present outside Ecdysozoa, for example in the anthozoan *Nematostella vectensis*, the echinoderm *Strongylocentrotus purpuratus* as well as in annelids and mollusks (*Jékely, 2013*; *Conzelmann et al., 2013*; *Stewart et al., 2014*; *Ahn et al., 2017*; *Zhang et al., 2018*). Our work confirms the presence of the complete bursicon signalling system in all arthropod genomes analysed here and extends its distribution (receptor and/or ligand) to the hemichordate, nemertean, phoronid, rotifer and onychophoran phyla (*Figure 2B*, *Figure 2—figure supplement 1D*, *Figure 2—source data 5*). Interestingly, bursicon and its receptor rickets are absent in tardigrades, suggesting the loss of the bursicon peptidergic signalling in this lineage (*Figures 2B* and *4B*, *Figure 2—figure supplement 6*, *Figure 2—source data 5*). The latter findings are corroborated by independent proneuropeptide and peptide prohormone surveys in the tardigrade genomic and EST data that also failed to detect this ligand-receptor pair in different tardigrade species (*Christie et al., 2011*; *Koziol, 2018*). We did not identify the receptor in any deuterostome or non-arthropod ecdysozoan lineage (*Figure 2B*).

The PTTH, ETH, EH, CCAP and bursicon peptide signalling systems are lacking in the nematode *Caenorhabditis elegans* (cf. our study and, e.g., *Page et al., 2014*; *Lažetić and Fay, 2017*; *Figure 2B*, *Figure 2—figure supplement 1*). Additionally, other key moulting components, such as the ecdysteroid ecdysone (E), 20-hydroxyecdysone (20E), and various halloween gene products have also been reported absent from the *C. elegans* genome (*Frand et al., 2005*; *Schumann et al., 2018*). An extensive body of research on moulting in *C. elegans* suggests an entirely different molecular machinery controlling this behaviour in this free-living nematode (*Russel et al., 2011*; for review *Lažetić and Fay, 2017*).

Interestingly, however, E and 20E were identified in parasitic nematodes (*Cleator et al., 1987*; *Shea et al., 2004*) and, outside Ecdysozoa, in the platyhelminth *Monieza expansa*, the gastropod mollusks *Lymnaea stagnalis* and *Helix pomatia* as well as in the hirudinean annelid *Hirudo medicinalis* (*Mendis et al., 1984*; *Nolte et al., 1986*; *Garcia et al., 1989*; *Barker et al., 1990*).

## Conclusion

We show that key peptidergic components of the arthropod ecdysis pathway emerged prior to the protostome-deuterostome split, and thus considerably earlier than commonly assumed. EH, CCAP and the bursicon signalling systems are more widespread among non-moulting animals than previously appreciated. The presence of the *eth-receptor* ortholog in ecdysozoans, lophotrochozoans and deuterostomes, in combination with the restriction of its known ligand to insects, arachnids and tardigrades, suggests a scenario in which promiscuous ligand/receptor relationships can lead to novel signalling interactions that provide new opportunities for natural selection to generate novel functions (*Figure 2B*). The identification of the near complete suite of the peptidergic arthropod ecdysis pathway components in Onychophora and Tardigrada strongly suggests that the entire pathway was at least functional in the last common ancestor of Panarthropoda and maybe as early as in the ur-ecdysozoan (*Figures 2B* and *4B*). However,

considering the crucial role of the ETH and bursicon signalling systems in insect moulting, together with the apparent secondary loss of ETH in Onychophora and bursicon in Tardigrada (*Figure 4B*), the consequences of harbouring only the partial set of the ecdysis signalling genes should be the focus of future assessments. Independent recruitment of novel peptidergic components into insect ecdysis has been shown (cf. *Kim et al., 2004*; *Kim et al., 2006a*; *Kim et al., 2006b*; extensively reviewed by *Zitnan et al., 1996*), illustrating the evolutionary plasticity of this signalling pathway and calling for more detailed functional investigations into the role of individual components during moulting of the various ecdysozoan lineages.

## Materials and methods

### Data collection, filtering, sequence reconstruction and proteome prediction

To obtain a comprehensive sampling across Metazoa, ecdysozoan, deuterostome and non-bilaterian protein-coding sequence (CDS) databases were downloaded from publicly available sites and combined with previous lophotrochozoan transcriptomes (see *De Oliveira et al., 2019*). The acoel transcriptomic data were pre-processed and assembled as described in *De Oliveira et al. (2019)*. The databases include representatives from the following phyla: Porifera, Ctenophora, Cnidaria, Placozoa, Xenacoelomorpha, Echinodermata, Hemichordata, Chordata, Annelida, Brachiopoda, Ectoprocta, Entoprocta, Gastrotricha, Mollusca, Nemertea, Phoronida, Platyhelminthes, Rotifera, Arthropoda, Tardigrada, Onychophora and Nematoda. The choanoflagellate *Monosiga brevicollis* was used as outgroup. *Supplementary file 1* summarises the databases and the publicly available repositories from which they were obtained. Sequence read archive (SRA) accession numbers for xenacoelomorph databases are also shown.

### Sensitive similarity searches with jackhmmer

Sensitive probabilistic iterative similarity searches based on profile hidden Markov models (HMMs) were performed with jackhmmer (*Johnson et al., 2010*) against the respective metazoan and choanoflagellate databases. Insect *eh*, *eth ccap*, *ptth* and *bursicon* orthologs were retrieved from NCBI (National Center for Biotechnology Information) and their respective receptors from *Vogel et al. (2013)*. These sequences were used as queries in the similarity searches. The searches were performed under the default parameters using varying e-value thresholds (1 to 1e-06) controlled by the options –E and –domE, as defined in jackhmmer. The best hits found in the metazoan and choanoflagellate databases were stored in fasta format and used in the subsequent analyses.

### Clustering and phylogenetic analyses

EH, ETH, CCAP, PTTH and bursicon ligand candidates retrieved from the metazoan and choanoflagellate databases were used as input, together with their respective insect orthologs, in the program clans (*Frickey and Lupas, 2004*) under different e-value thresholds (0.1 to 1e-06) and blast programs, that is blastp or psiblast (*Camacho et al., 2009*). Singleton sequences (isolated unconnected sequences) were excluded from the map. To further improve the orthology assessment, multiple sequence alignments were performed with mafft (*Katoh and Standley, 2013*) and the presence of shared conserved amino acid regions and residues were investigated with aliview (*Larsson, 2014*). The final 3D maps were collapsed into 2D after the clustering for easier visualisation.

Putative EH, ETH, CCAP, PTTH and bursicon receptor candidates retrieved from the metazoan and choanoflagellate databases were aligned with mafft together with their respective orthologs, when found, and subsequently trimmed with BMGE software under the following parameters: –h 1 –b 1 –m BLOSUM30 –t AA (*Criscuolo and Gribaldo, 2010*). Outgroups for the phylogenetic analyses were defined according to *Vogel et al. (2013)*.

Phylogenetic analyses were performed using RAxML (*Stamatakis, 2014*), PhyML (*Guindon et al., 2010*) and mrbayes (*Ronquist et al., 2012*) softwares using the appropriate best-fit model of amino acid substitution. RaxML was executed under default parameters and rapid bootstrap. PhyML was executed under the default parameters and an optimised starting tree (-o tlr option). The number of bootstrap values was set to 1.000 in RaxML and PhyML and the number of generations used in mrbayes was determined using a convergence diagnostic. All runs in mrbayes were performed with

the samplefreq and relative burn-in defined as 1000 and 25%, respectively. The three final phylogenetic trees obtained for each of the four different receptors were visualised and combined with TreeGraph2 (*Stöver and Müller, 2010*).

## Data availability

All data generated in the course of this study are included in this article (*Figure 2—source datas 1–5* and *Figure 3—source data 1*). The accession numbers for the publicly available datasets used in this work are available in *Supplementary file 1*. The 3D cluster peptide maps can be visualised and manipulated using the program clans (*Frickey and Lupas, 2004*); see ftp://ftp.tuebingen.mpg.de/pub/protevo/CLANS/). The multiple sequence alignment files can be viewed with aliview (*Larsson, 2014*). The phylogenetic tree files can be viewed using Figtree (http://tree.bio.ed.ac.uk/software/figtree/) or TreeGraph2 (*Stöver and Müller, 2010*).

# Additional information

### Funding

| Funder | Grant reference number | Author |
|---|---|---|
| Austrian Science Fund | P29455-B29 | Andreas Wanninger |
| Coordenação de Aperfeiçoamento de Pessoal de Nível Superior | 6090/13-3 | André Luiz de Oliveira |

The funders had no role in study design, data collection and interpretation, or the decision to submit the work for publication.

### Author contributions

André Luiz de Oliveira, Conceptualization, Resources, Data curation, Software, Formal analysis, Funding acquisition, Validation, Investigation, Visualization, Methodology, Writing—original draft, Writing—review and editing; Andrew Calcino, Conceptualization, Methodology, Writing—review and editing; Andreas Wanninger, Conceptualization, Resources, Supervision, Funding acquisition, Investigation, Project administration, Writing—review and editing

### Author ORCIDs

André Luiz de Oliveira (iD) https://orcid.org/0000-0003-3542-4439
Andrew Calcino (iD) https://orcid.org/0000-0002-3956-1273
Andreas Wanninger (iD) https://orcid.org/0000-0002-3266-5838

### Decision letter and Author response

Decision letter https://doi.org/10.7554/eLife.46113.022
Author response https://doi.org/10.7554/eLife.46113.023

# Additional files

### Supplementary files

• Supplementary file 1. List of molecular databases included in this study. Superphylum and/or phylum of the investigated species and the online repositories for each of the databases are also listed.
DOI: https://doi.org/10.7554/eLife.46113.019
• Transparent reporting form
DOI: https://doi.org/10.7554/eLife.46113.020

### Data availability

All data generated or analysed during this study are included in the manuscript and supporting files. Source data files have also been provided. The molecular databases analysed in this study are publicly available and novel annotated sequences are included in the Source data 1-5. The list of

investigated species and their respective links to a direct download are presented in the Supplementary File 1 (Table S1). The 3D proneuropeptide/prohormone maps as well as all the multiple sequence alignments and the phylogenetic trees generated in this study are available in the Source Data 1-5 enclosed in the original submission. The 3D maps in .rtf format can be visualised and inspected with the software clans (ftp://ftp.tuebingen.mpg.de/pub/protevo/CLANS/). The multiple sequence alignments used in the phylogenetic inferences can be graphically visualised using aliview (http://www.ormbunkar.se/aliview/#DOWNLOAD). The phylogenetic tree files can be viewed using an appropriate phylogetic tree viewer such as Figtree (http://tree.bio.ed.ac.uk/software/figtree/).

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
