## [Decision Letter]

Thank you for submitting your article "Ancient origins of arthropod moulting pathway components" for consideration by *eLife*. Your article has been reviewed by K VijayRaghavan as the Reviewing Editor and Senior Editor, and two reviewers. The following individual involved in review of your submission has agreed to reveal his identity: Pedro Martinez (Reviewer #1).

The reviewers have discussed the reviews with one another and the Reviewing Editor has drafted this decision to help you prepare a revised submission.

The paper by De Oliveira and collaborators focus on a very interesting evolutionary problem, the origin of animal "moulting". The main finding of the study is that the presence of the molecular machinery involved in the moulting process (within the Ecdysozoa) is also present in many bilaterian clades. This is another example (we have seen many over the last years) of what has been called "preadaptation", in which the genesis of the molecular components of a specific process predates its overt manifestation.

The paper summarizes an enormous amount of bioinformatics analysis of genomes and transcriptomes present in public databases. De Oliveira and collaborators have identified some putative components of the moulting process in different animal groups and have assessed their true homologies through extensive, rigorous, phylogenetic analysis. The analysis is comprehensive and the conclusions are both sound and relevant. Moreover, the manuscript is very well written, clear, and avoids entering into unnecessary speculation.

The above seems like a rather straightforward conclusion of what is present in the paper though the findings in themselves still provide us with an incomplete picture of what has really happened over evolutionary time. Having the "components" is not equivalent to using the "process" (moulting). It is obvious, though, that without the preliminary identification of components it is impossible to test for functionality.

Yet, we have several important concerns that require careful and well-argued re-writing if this data is to be distilled and understood for its strengths and limitations.

Essential revisions:

1) In absence of gene expression data it is impossible to know whether these genes are used for anything reminiscent of a moulting process. It would be much better if we could have the expression profiles of those genes (in space and time) for any clade, outside the Arthropoda. This will help us understanding the roles that these genes/components have in other animals. This may not be readily experimentally feasible, but if some data is available, this could be speedily determined.

2) Co-option is a rather common mechanism in metazoan development. The interesting aspect of it is to understand how they are co-opted in different clades and for what role (s). Are all of them co-opted "en bloc"? (or not) and what are the different consequences of having the whole set of "moulting" components or just a fraction of them? Are they used for similar functions in different clades? Needless to say, functional data (knockdown) will clarify some of the processes in which these orthologous genes are involved. Again, we do not expect the authors to attend to this is in this study, but this can be highlighted in the Discussion section.

3) Why do the authors select these particular genes? Obviously the moulting process involves a series of complex molecular cascades of which the pair hormone/receptor is a minimal part. What about the downstream components?

4) At the end we are left with a comprehensive (and particularly relevant) analysis of the evolutionary history of the molecular components involved in arthropod moulting but also without a clear intuition of what they might be used for outside this phylum. This gap also needs to be highlighted in the writing.

5) It is already known that many of the peptides and receptors involved in the control of molting and ecdysis are members of ancient families of ligands and receptors. For example, the ETH receptors are members of a large family of GPCRs, the neuromedin U group of receptors (Park., et al., 2003; also Park et al., 2002) that include the vertebrate receptors for thyrotropin-releasing hormone and neuromedin U. The members of this family in the insects include receptors for the cardioactive peptide CAP2b, the pyrokinins, and the ETHs. The receptors and ligands are similar enough that there is considerable promiscuity amongst the members, a relationship consistent with coevolution of the ligands and their receptors. I guess that the basic question is that as one examines the emergence of a ligand and receptor from a more ancient family, what criteria does one use to identify when the ecdysis-related ligand/receptor has been established? It is difficult to pull this information out of the trees in the supplemental information. This issue needs to be squarely addressed by substantial re-writing.

6) A case in point is PTTH. The abstract states that "Prothoracicotropic hormone, a neuropeptide that triggers ecdysis in insects and paralog of the ancient signalling peptide trunk, is present in the pre-bilaterian ctenophore Mnemiopsis leidyi. " However, as the authors state early in the paper, PTTH has not been found outside of the insects. The gene data on the occurrence of PTTH in accord with the physiology of control of ecdysone secretion from the Y-organ/prothoracic glands in the crustaceans/insects. Secretion is under positive control by PTTH [and also insulin-like peptides] in insects but under negative control by molt-inhibiting hormone in the crustaceans. It is hard to believe that PTTH could have arisen prior to the evolution of Ctenophores and maintained in that group, while it was being systematically lost in all of the metazoans until was retained in the insects. The clustering in Figure 3 shows the Ctenophore protein is quite distant from both the trunk and PTTH clusters. How it can be claimed to be a PTTH homolog?

[Editors' note: further revisions were requested prior to acceptance, as described below.]

Thank you for resubmitting your article "Ancient origins of arthropod moulting pathway components" for consideration by *eLife*. Your article has been re-reviewed by K VijayRaghavan as the Senior Editor and Reviewing Editor, and two reviewers. The following individual involved in review of your submission has agreed to reveal his identity: Pedro Martinez (Reviewer #1).

The reviewers have discussed the reviews with one another and the Reviewing Editor has drafted this decision to help you prepare a revised submission.

Summary:

De Oliveira and collaborators have sent a revised, and much, improved version of the manuscript. Though the main findings are the same, the way the data is presented is much better. Some illustrations, such as Figure 2, synthesize clearly the results obtained.

As mentioned in the previous review, the phylogenetic analysis is solid and very well performed. This is a very thorough analysis of the moulting components across the metazoan, using a wide taxonomic range of animal genomes and transcriptomes. This deserves to be published in the current state, but some points need to be addressed, which hopefully can be quickly done; the separate reviews point to these.

Essential revisions:

*Reviewer #1:*

Despite the fact that the data is solid and the text clearly summarizes the (huge) amount of phylogenetic data, I think it is important to point out a few things relevant to the study (some of the points recognized by the authors):

1) The lack of a clearly "moulting-specific" phenotypes generated in *Drosophila melanogaster* when peptides such as PTTH, EH (in some studies), CCAP, etc. are inactivated shows clearly that the role of these peptide systems is not restricted to the moulting process. The co-option of these molecules from other, more generalized, functions (i.e. neural roles, reproduction or regulation of metabolic processes) is demonstrated by the phylogenetic distribution of these peptides. Thus, these peptides have different roles, many, most probably, unrelated to moulting.

2) The loss of the CCAP ligand in deuterostomes, while they retain its receptor, is puzzling. Do this receptor binds other ligands? (I assume it does). Conversely, the ligand bursicon is present in hemichordates, while the receptor is missing.

Are those two examples suggesting promiscuity of those systems in bilaterian groups or just methodological problems in the identification process? (I.e. qualities of the genomes/transcriptomes).

My impression is that the promiscuity of the systems is a salient feature that needs to be considered (the authors are aware of it).

3) Moreover, the presence of steroids such as ecdysone and 20-hydroxyecdysone in phyla such as Nematoda, Platyhelminthes, Annelida or Mollusca is quite striking, giving the fact that these are components well recognized as (canonical) initiators of the arthropod moulting process. What do these steroids control here?

All the above comments suggest a wide spectrum of functions carried out by these systems in the metazoans. Co-option for moulting in arthropods and, probably, in Ecdysozoa seems quite clear. I find striking the wide distribution of these peptide systems in most metazoan phyla. In some of those, a full complement of "moulting" peptide/receptor components seems present. A series of interrelated questions arise: (1) what do they do there? Are all functions unrelated to moulting? If that is the case, (2) how the arthropod moulting machinery is assembled? Is that something that happens "at once" in evolutionary time? If that were the case, (3) how come that in other phyla where all or most components are present we don't see any overt sign of moulting? This, as suggested before, would mean that the regulatory cross-reactions between all peptide systems happen quite abruptly! and, of course, (4) what is special about insect moulting that makes the process so obvious there but nowhere else?

It is clear to me that all these questions are particularly relevant and, obviously, we wouldn't be able to formulate them in absence of a thorough analysis of the distribution of components among different animal phyla. What is presented here is a necessary first step to understand the assembly of the moulting program. But not just this, it opens the possibility of analysing how these different components are used in different groups and for what specific purposes. This analysis should provide us with insights as to how the moulting process is established over evolutionary time, and, perhaps, will allow us to understand whether moulting comes in many different guises.

In the Abstract the authors say in the last sentence "This constitutes the first ctenophore signalling peptide with homology to a part of the bilaterian neuropeptide complement". The sentence is not very clear. Do they mean that there is one peptide family that is present in ctenophores? I guess the sentence can be written more clearly.

Giving the conflicting topologies given by different authors to the interrelationships between the major metazoan phyla, I would suggest the authors to give the reference they are using in Figure 2.

*Reviewer #2:*

The de Oliveira et al. paper has been improved since its original submission. Although ecdysis was originally thought to be a highly specialized behavior for arthropods, this analysis shows that components of the ecdysis control system can be found through the Ecdysozoa and may extend down to the base of the Bilateria. These appear to be a potentially ancient control systems and we need to understand their ancestral functions. The paper provides an important first step in this direction.

The one objection that I have to the paper is their statement starting on line 98 that the relationship of PTTH to a ctenophore peptide "constitutes the first homologous relationship between a ctenophore signalling peptide and a bilaterian neuropeptide." This could be an important conclusion because of the current controversy as to whether the Ctenophores independently evolved neurons and a nervous system. PTTH is a member of the Trunk signaling family. These are important signaling molecules in early embryonic development but PTTH is unusual in that it is the only family member that is a neuropeptide. The PTTH function was thought to have been evolved within the insects and the authors' analysis support this conclusion. Considering trunk's role in early development of metazoans, I am not surprised that Ctenophores have a member of this family. I see no justification, though, to call it a PTTH homolog. This statement should also be struck from the Abstract.

---

## [Author Response]

Essential revisions:1) In absence of gene expression data it is impossible to know whether these genes are used for anything reminiscent of a moulting process. It would be much better if we could have the expression profiles of those genes (in space and time) for any clade, outside the Arthropoda. This will help us understanding the roles that these genes/components have in other animals. This may not be readily experimentally feasible, but if some data is available, this could be speedily determined.

We agree with the reviewers’ comment. Comparative studies on the temporal and spatial gene expression patterns of important components of the moulting pathways across the metazoan phyla represent an initial step to infer the ancestral and novel functions of these genes. Unfortunately, however, the majority of the gene expression studies on the neuroendocrine components of the moulting pathway are currently restricted to a very limited number of arthropods, and functional analyses (e.g. RNAi, gene knockouts) only exist for a few so-called arthropod “model organisms” (e.g. *Drosophila melanogaster, Tribolium castaneum*). Therefore, robust experimental evidence for the putative function of the moulting genes in our prime target animals, polyplacophorans, scaphopods, and bivalves, is yet impossible (and were not the main subject of this study).

Recognizing the importance of a comparative survey as mentioned by the reviewer, we conducted a thorough bibliographic survey and found three interesting studies on the crustacean cardioactive peptide (CCAP) signalling pathway in mollusks (Vehovszky et al., 2005; In et al., 2016; Endress et al., 2018). These showed a dense network of CCAP-positive fibers in the buccal neural mass of the snail *Lymnaea stagnalis* (Vehovszky et al., 2005), likely regulating part of the feeding behaviour in the gastropod. In the oyster *Saccostrea glomerata* (In et al., 2016) and the cuttlefish *Sepia officinalis (*Endress et al., 2018) in vivo bioassays and immunohistochemistry suggested that the CCAP signalling pathway is involved in reproduction (e.g. spawning, oocyte transport). Additionally, in the cephalopod, CCAP has been shown to increase the tonus of the vena cava demonstrating its role in the regulation of hemolymph circulation.

These results are particularly interesting when compared to the “non-moulting” role of CCAP in arthropods. As described in the manuscript, CCAP accelerates the frequency of oviduct contractions in the locust *Locusta migratori*a,and regulates the release of digestive enzymes in the cockroach *Periplaneta americana*. The results indicate that in both mollusks and arthropods, CCAP functions in feeding, reproduction and regulation of hemolymph circulation, suggesting that these may have been its ancestral roles (as also discussed previously by Endress et al., 2018). In arthropods, co-option of CCAP in to the ecdysis pathway expanded this set of functions to include moulting.

A paragraph discussing the role of CCAP outside of Arthropoda was included in the main manuscript Results and Discussion section together with new bibliographic references (Vehovszky et al., 2005; Endress et al., 2018; In et al., 2016).

2) Co-option is a rather common mechanism in metazoan development. The interesting aspect of it is to understand how they are co-opted in different clades and for what role (s). Are all of them co-opted "en bloc"? (or not) and what are the different consequences of having the whole set of "moulting" components or just a fraction of them? Are they used for similar functions in different clades? Needless to say, functional data (knockdown) will clarify some of the processes in which these orthologous genes are involved. Again, we do not expect the authors to attend to this is in this study, but this can be highlighted in the Discussion section.

As suggested by the reviewers we included a paragraph in the manuscript highlighting the reviewers’ points discussed above. The newly incorporated section in the manuscript can be found in the main manuscript (Conclusion section):

“However, considering the crucial role of the ETH and bursicon signalling systems in insect moulting, together with the apparent secondary loss of ETH in Onychophora and bursicon in Tardigrada (Figure 4B), the consequences of harboring only the partial set of the ecdysis signalling genes should be the focus of future assessment.”

Furthermore, we would like to address here the “en bloc” question raised by the reviewers. As there are almost no data available on gene expression on the ecdysis genes outside of Arthropoda, we can only speculate about a possible cascade of gene expression patterns in non-bilaterian, lophotrochozoan and deuterostome animals, and our answer is “not likely”. From studies within arthropods, it is clear that ecdysis is a complex of coordinated behaviours underlain by well-defined cascades of ecdysteroid and peptide signalling elements. Despite “moulting” is classified as a defining character of ecdysozoans, the majority of its molecular repertoire predates the Ecdysozoa clade itself, showing that this signalling pathway is built upon already pre-existing ligand and receptor molecules. In addition, the process of moulting and the molecules involved vary widely between ecdysozoan clades, e.g. between insects and nematodes, an issue we also address in the manuscript.

3) Why do the authors select these particular genes? Obviously the moulting process involves a series of complex molecular cascades of which the pair hormone/receptor is a minimal part. What about the downstream components?

As pointed out by the reviewers, the neuroendocrine basis for moulting encompasses many cellular and chemical signalling mechanisms, which includes ecdysteroids, peptide hormones, and neuropeptides. As a broad comparative *in-silico* analysis has been recently published (Schumann et al., 2018) elucidating the evolution and distribution of the genes responsible for ecdysteroid production and its biosynthesis, we focused our efforts on the peptidergic signalling pathways (i.e. neuropeptides and peptide hormones). By taking into consideration the different proposed models for ecdysis behaviour control in different insects (e.g. *Drosophila melanogaster, Manduca sexta, Tribolium castaneum),* we selected five main ligand-receptor components present in the three distinct stages of ecdysis (pre-ecdysis, ecdysis and post-ecdysis): p*rothoracicotropic* hormone, ecdysis-triggering hormone, eclosion hormone, crustacean cardioactive peptide, and bursicon. The classification of these peptide hormones and neuropeptides as “key” or “main” is not unique to us, and several studies have been published recognising these elements as major players in the moulting behavior (as referenced in our work; e.g., Arakane et al., 2008; Zitnan and Adams, 2012). We acknowledged in the manuscript that differences in the peptidergic components among different insect do exist. The respective paragraph reads:

“Independent recruitment of novel peptidergic components into insect ecdysis has been shown (cf. Kim et al., 2004, 2006a,b; extensively reviewed by Zitnan & Adams, 2012) illustrating the evolutionary plasticity of this signalling pathway and calls for more detailed functional investigations into the role of individual components during moulting of the various ecdysozoan lineages.”

To conclude, the core set of peptidergic signalling components investigated here represent the most conserved components of the arthropod ecdysis signalling pathway.

4) At the end we are left with a comprehensive (and particularly relevant) analysis of the evolutionary history of the molecular components involved in arthropod moulting but also without a clear intuition of what they might be used for outside this phylum. This gap also needs to be highlighted in the writing.

We have included in the manuscript one paragraph briefly discussing the putative function of the CCAP signalling pathway in mollusks, suggesting a hypothetical evolutionary scenario for the function of these ligand-receptor genes in prostostomian and ecdysozoan animals (please see also our answer for question 1). The respective paragraph reads as follows (Results and Discussion section):

“Only three studies focusing on the CCAP signalling pathway components are available outside of Arthropoda. In the snail *Lymnaea stagnalis* (Vehovszky et al., 2005), immunostaining revealed a dense network of CCAP-positive fibers that likely function to regulate parts of the feeding behaviour. In the oyster *Saccostrea glomerata* (In et al., 2016) and in the cuttlefish *Sepia officinalis* (Endress et al., 2018), in vivo bioassays using synthesized neuropeptides and immunohistochemistry suggested that the CCAP signalling pathway is involved in reproduction (e.g., spawning, oocyte transport, egg-laying). Additionally, *Sepia* CCAP has been shown to increase the tonus of the vena cava, demonstrating its role in the regulation of hemolymph circulation (Endress et al., 2018). These results indicate that in both, mollusks and arthropods, CCAP functions in feeding, reproduction, and regulation of hemolymph circulation, suggesting that these may have been its ancestral roles. In arthropods, co-option of CCAP in to the ecdysis pathway expanded this set of functions to include moulting.”

To the best of our knowledge, there is no data available outside of Arthropoda on the remaining moulting components investigated in this work, i.e. trunk, ecdysis-triggering hormone, eclosion hormone, and bursicon. It is likely that each of these components have varying and independent functions from one another outside the ecdysozoa (e.g., feeding behaviour, reproduction and hemolymph circulation by CCAP) and that their co-option in to the arthropod ecdysis pathway represents a unique circumstance in which their functions were coordinated for a particular physiological process.

5) It is already known that many of the peptides and receptors involved in the control of molting and ecdysis are members of ancient families of ligands and receptors. For example, the ETH receptors are members of a large family of GPCRs, the neuromedin U group of receptors (Park., et al., 2003; also Park et al., 2002) that include the vertebrate receptors for thyrotropin-releasing hormone and neuromedin U. The members of this family in the insects include receptors for the cardioactive peptide CAP2b, the pyrokinins, and the ETHs. The receptors and ligands are similar enough that there is considerable promiscuity amongst the members, a relationship consistent with coevolution of the ligands and their receptors. I guess that the basic question is that as one examines the emergence of a ligand and receptor from a more ancient family, what criteria does one use to identify when the ecdysis-related ligand/receptor has been established? It is difficult to pull this information out of the trees in the supplemental information. This issue needs to be squarely addressed by substantial re-writing.

In order to make clearer the evolutionary history of the ETH signalling system amongst different metazoan phyla and within Arthropoda, we incorporated into the manuscript a more fined-grained overview of the evolution of ETH receptor and ligand. The respective paragraph read as follows (Results and Discussion section):

“For Arthropoda, the ETH ligand was only found in insects (*Drosophila* and *Tribolium)* but was lacking in the crustacean *Parhyale hawaiensis* and the arachnid *Parasteatoda tepidariorum*.However, studies on the two mite species *Panonychus citri* and *Tetranychus urticae* have shown the presence of the ETH ligand in these chelicerates (in which the homology was reconfirmed by our clustering analysis) (Veenstra et al., 2012; Zhu et al., 2019).To date, no publicly available ETH ligand gene has been reported in crustaceans.”

Concerning the reviewers’ question, we are in agreement with them. It is difficult (even impossible) to pinpoint the establishment of a signalling system based solely on phylogenetic analyses and the presence or not of specific genes in different clades. In order to conclusively say anything about up/down-stream effects and roles of specific genes (e.g. cell responses induced by a receptor), one must combine pharmaceutical and functional screening with gene expression data. The elucidation of a signalling pathway is not a trivial task and was not the focus of this work.

The claim that the ecdysis-triggering hormone ligand-receptor pair has been established in the ecdysis pathway derives from the many functional studies on these genes in different insects (as discussed in the Results and Discussion section), rather than the results of our phylogenetic analyses and clustering approaches. Our study only shows that distribution of the ETH receptor is widespread in many lophotrochozoan and deuterostome clades, and that its ligand is found within Panarthropoda. While the ligand for the non-ecdysozoan ETH receptor homologs is not known, our results suggest the ETH ligand/receptor relationship emerged at the base of the ecdysozoans. This ligand/receptor innovation is an example of the promiscuity mentioned by the reviewers. The following sentence in the manuscript aims to describe this scenario:

“The presence of the *eth-receptor* ortholog in ecdysozoans, lophotrochozoans and deuterostomes, in combination with the restriction of its known ligand to insects, arachnids and tardigrades, suggests a scenario in which promiscuous ligand/receptor relationships can lead to novel signalling interactions that provide new opportunities for natural selection to generate novel functions (Figure 2B).”

6) A case in point is PTTH. The abstract states that "Prothoracicotropic hormone, a neuropeptide that triggers ecdysis in insects and paralog of the ancient signalling peptide trunk, is present in the pre-bilaterian ctenophore Mnemiopsis leidyi. " However, as the authors state early in the paper, PTTH has not been found outside of the insects. The gene data on the occurrence of PTTH in accord with the physiology of control of ecdysone secretion from the Y-organ/prothoracic glands in the crustaceans/insects. Secretion is under positive control by PTTH [and also insulin-like peptides] in insects but under negative control by molt-inhibiting hormone in the crustaceans. It is hard to believe that PTTH could have arisen prior to the evolution of Ctenophores and maintained in that group, while it was being systematically lost in all of the metazoans until was retained in the insects. The clustering in Figure 3 shows the Ctenophore protein is quite distant from both the trunk and PTTH clusters. How it can be claimed to be a PTTH homolog?

We thank the reviewers for pointing out an error in our Abstract. As mentioned by them, our analyses show that the phylogenetic distribution of prothoracicotropic hormone ligand, PTTH, is so far restricted to insects. We rephrased the Results and Discussion section to read:

“By screening 39 metazoan genomes and 57 transcriptomes (Supplementary file 1; De Oliveira et al., 2019), we found that the PTTH peptide is present in *Drosophila* and *Tribolium* but absent in the house spider *Parasteatoda tepidariorum* and the crustacean *Parhyale hawaiensis*), suggesting that PTTH is an insect innovation.”

To fix this issue and adjust the Abstract to the correct findings of our study, the original erroneous sentence has been rewritten. Additionally, a more careful writing of these results in the Results and Discussion section has been added. See text in Abstract: “This constitutes the first ctenophore signalling peptide with homology to a part of the bilaterian neuropeptide complement.” And in the Results and Discussion section of the revised manuscript version:

“By similarity-based clustering we were also able to demonstrate homology of the ctenophore trunk-like peptide with the insect *trunk* paralog, *ptth* (Figure 3A), which constitutes the first homologous relationship between a ctenophore signalling peptide and a bilaterian neuropeptide (see, e.g., Halanych, 2004; Dunn et al., 2008; Moroz et al., 2014; Jékely et al., 2015; Pisani et al., 2015 for discussion).”

Regarding the hypothesis that the comb jelly trunk peptide is homologous to the insect PTTH, we believe that the evidence gathered from our results associated with the current information available in the scientific literature support this conclusion. We cannot definitely rule out convergent evolution; however, this hypothesis is not supported by the evidence. Conversely, a robust body of data supports our claim:

1) PTTH and trunk are closely related (even sharing the same putative tyrosine kinase receptor named torso); with PTTH being the closest paralog of trunk (Figure 2—figure supplement 1, Rewitz et al., 2009);

2)Trunk has been previously described outside of Arthropoda, in protostome and deuterostome animals (Jékely, 2013), and thus hypothesized to be present already in the Urbilateria;

3) This study identifies both the trunk ligand and its putative receptor (i.e. torso) in non-bilaterian animals (i.e. cnidarians and one ctenophore), as shown by three different methods: clustering (Figure 3A), Bayesian and maximum-likelihood phylogenetic analyses (Figure 2—figure supplement 2);

4) *Mnemiopsis leidyi s*equence is connected to lophotrochozoan, deuterostome and ecdysozoan trunk sequences with good p-values (<1e-05) (Figure 3A). Additionally, all these sequences share the key diagnostic conserved cysteine amino acid residues present in trunk peptides (Figure 3B – please check the conservation histogram);

5) The clustering approach has been proven to be a powerful method to reveal homology between distantly related sequences, through the identification of indirect links in the network of BLAST interactions (Jékely, 2013; Conzelmann et al., 2013; De Oliveira et al., 2019).

Based on the aforementioned evidences, we proposed a scenario in which the trunk/torso signalling pathway was already present in the last common eumetazoan ancestor (all animals expect sponges) with a duplication of trunk at the base of Insecta, giving rise to the paralog PTTH and to the trunk/torso/ptth signalling pathway. We would like to stress that we are not claiming that the ctenophore sequence is a proneuropeptide or peptide prohormone, but rather, that this finding constitutes the first ctenophore signalling peptide with homology to a part of a bilaterian neuropeptide complement (i.e. trunk/PTTH/torso).

[Editors' note: further revisions were requested prior to acceptance, as described below.]

Essential revisions:

Reviewer #1:

Despite the fact that the data is solid and the text clearly summarizes the (huge) amount of phylogenetic data, I think it is important to point out a few things relevant to the study (some of the points recognized by the authors):1) The lack of a clearly "moulting-specific" phenotypes generated in Drosophila melanogaster when peptides such as PTTH, EH (in some studies), CCAP, etc. are inactivated shows clearly that the role of these peptide systems is not restricted to the moulting process. The co-option of these molecules from other, more generalized, functions (i.e. neural roles, reproduction or regulation of metabolic processes) is demonstrated by the phylogenetic distribution of these peptides. Thus, these peptides have different roles, many, most probably, unrelated to moulting.

We agree with the reviewer – exactly such a statement is provided in the Conclusion part of our manuscript.

2) The loss of the CCAP ligand in deuterostomes, while they retain its receptor, is puzzling. Do this receptor binds other ligands? (I assume it does). Conversely, the ligand bursicon is present in hemichordates, while the receptor is missing.Are those two examples suggesting promiscuity of those systems in bilaterian groups or just methodological problems in the identification process? (I.e. qualities of the genomes/transcriptomes).My impression is that the promiscuity of the systems is a salient feature that needs to be considered (the authors are aware of it)

This is indeed a very interesting point. Obviously, such studies are way beyond the scope of this work, but these are clearly questions that should be addressed in the future. Our study points towards these questions by helping to formulate functional hypotheses that can (and should) now be tested by using state-of-the-art functional approaches.

3) Moreover, the presence of steroids such as ecdysone and 20-hydroxyecdysone in phyla such as Nematoda, Platyhelminthes, Annelida or Mollusca is quite striking, giving the fact that these are components well recognized as (canonical) initiators of the arthropod moulting process. What do these steroids control here?

Another very interesting point that is the result of our study and that raises such a functional question. As with the issue raised above, this should now be tested using functional genetic approaches.

All the above comments suggest a wide spectrum of functions carried out by these systems in the metazoans. Co-option for moulting in arthropods and, probably, in Ecdysozoa seems quite clear. I find striking the wide distribution of these peptide systems in most metazoan phyla. In some of those, a full complement of "moulting" peptide/receptor components seems present. A series of interrelated questions arise: (1) what do they do there? Are all functions unrelated to moulting? If that is the case, (2) how the arthropod moulting machinery is assembled? Is that something that happens "at once" in evolutionary time? If that were the case, (3) how come that in other phyla where all or most components are present we don't see any overt sign of moulting? This, as suggested before, would mean that the regulatory cross-reactions between all peptide systems happen quite abruptly! and, of course, (4) what is special about insect moulting that makes the process so obvious there but nowhere else?

We couldn’t agree more. We are delighted to read that this reviewer already developed important and interesting questions based on our work. Clearly, these can now be specifically addressed. And we are glad if our paper will inspire other researchers as well to tackle these issues.

It is clear to me that all these questions are particularly relevant and, obviously, we wouldn't be able to formulate them in absence of a thorough analysis of the distribution of components among different animal phyla. What is presented here is a necessary first step to understand the assembly of the moulting program. But not just this, it opens the possibility of analysing how these different components are used in different groups and for what specific purposes. This analysis should provide us with insights as to how the moulting process is established over evolutionary time, and, perhaps, will allow us to understand whether moulting comes in many different guises.Thank you, this is exactly the point of our work.A minor point: in the Abstract the authors say in the last sentence "This constitutes the first ctenophore signalling peptide with homology to a part of the bilaterian neuropeptide complement". The sentence is not very clear. Do they mean that there is one peptide family that is present in ctenophores? I guess the sentence can be written more clearly.

We rephrased this sentence accordingly. It now reads “Trunk, an ancient extracellular signalling molecule and a well-established paralog of the insect peptide prothoracicotropic hormone (PTTH), is present in the non-bilaterian ctenophore *Mnemiopsis leidyi*. This constitutes the first case of a ctenophore signalling peptide with homology to a neuropeptide.” We simplified the last part of the sentence to hopefully make the sentence easier to read.

Giving the conflicting topologies given by different authors to the interrelationships between the major metazoan phyla, I would suggest the authors to give the reference they are using in Figure 2.

Done. Reference was already given for 2B, now we added it also for 2A, including a statement that some taxa are omitted in the simplified figure for clarity.

Reviewer #2:

The de Oliveira et al. paper has been improved since its original submission. Although ecdysis was originally thought to be a highly specialized behavior for arthropods, this analysis shows that components of the ecdysis control system can be found through the Ecdysozoa and may extend down to the base of the Bilateria. These appear to be a potentially ancient control systems and we need to understand their ancestral functions. The paper provides an important first step in this direction.The one objection that I have to the paper is their statement starting on line 98 that the relationship of PTTH to a ctenophore peptide "constitutes the first homologous relationship between a ctenophore signalling peptide and a bilaterian neuropeptide." This could be an important conclusion because of the current controversy as to whether the Ctenophores independently evolved neurons and a nervous system. PTTH is a member of the Trunk signaling family. These are important signaling molecules in early embryonic development but PTTH is unusual in that it is the only family member that is a neuropeptide. The PTTH function was thought to have been evolved within the insects and the authors' analysis support this conclusion. Considering trunk's role in early development of metazoans, I am not surprised that Ctenophores have a member of this family. I see no justification, though, to call it a PTTH homolog. This statement should also be struck from the Abstract.

We want to stress here again that the finding/suggestion that PTTH and trunk are homologs (in fact, paralogs), is not ours, but the results of previous works and is now well established (Rewitz et al., 2009; Jékely, 2013). Our work only extends the phyletic distribution of Trunk to ctenophores and recognises that this represents the first case of homology between a ctenophore signalling peptide and a bilaterian neuropeptide. To make the main text clearer in this respect, we partially rephrased the first paragraph of the Results and Discussion section. It now reads: “By similarity-based clustering we were able to demonstrate homology of the ctenophore trunk-like peptide with the insect *trunk* paralog, *ptth* (Figure 3A, Figure 3—source data 1; see also Rewitz et al., 2009; Jékely, 2013). This extends the phyletic distribution of trunk to the ctenophores (see, e.g., Halanych, 2004; Dunn et al., 2008; Moroz et al., 2014; Jékely et al., 2015; Pisani et al., 2015 for discussion).

Our finding that trunk is present in ctenophores has no impact on the competing hypotheses on the evolutionary origins of the nervous system. As pointed out by Jékely et al., (2015): “Even if neuropeptides and their receptors are homologous, their presence is not sufficient evidence for nervous system homology since *Trichoplax*, an animal that lacks a morphologically recognizable nervous system, also possesses these molecules.”

See also quote from Jékely, (2013): “The arthropod PTTHs are related to the extracellular signaling molecule trunk (69) that is a member of an ancient bilaterian family; trunk orthologs could be identified in annelids, mollusks, and in *B. floridae*.”